# From Yeast to Mammals, the Nonsense-Mediated mRNA Decay as a Master Regulator of Long Non-Coding RNAs Functional Trajectory

**DOI:** 10.3390/ncrna7030044

**Published:** 2021-07-27

**Authors:** Sara Andjus, Antonin Morillon, Maxime Wery

**Affiliations:** 1ncRNA, Epigenetic and Genome Fluidity, Institut Curie, PSL University, Sorbonne Université, CNRS UMR3244, 26 Rue d’Ulm, CEDEX 05, F-75248 Paris, France; sara.andus@curie.fr; 2ncRNA, Epigenetic and Genome Fluidity, Institut Curie, Sorbonne Université, CNRS UMR3244, 26 Rue d’Ulm, CEDEX 05, F-75248 Paris, France

**Keywords:** nonsense-mediated mRNA decay, Upf1, lncRNA, translation, micropeptide

## Abstract

The Nonsense-Mediated mRNA Decay (NMD) has been classically viewed as a translation-dependent RNA surveillance pathway degrading aberrant mRNAs containing premature stop codons. However, it is now clear that mRNA quality control represents only one face of the multiple functions of NMD. Indeed, NMD also regulates the physiological expression of normal mRNAs, and more surprisingly, of long non-coding (lnc)RNAs. Here, we review the different mechanisms of NMD activation in yeast and mammals, and we discuss the molecular bases of the NMD sensitivity of lncRNAs, considering the functional roles of NMD and of translation in the metabolism of these transcripts. In this regard, we describe several examples of functional micropeptides produced from lncRNAs. We propose that translation and NMD provide potent means to regulate the expression of lncRNAs, which might be critical for the cell to respond to environmental changes.

## 1. Introduction

The accurate transmission of the genetic information is crucial for the cell, and several surveillance mechanisms have evolved to monitor the distinct steps of gene expression. RNA surveillance pathways are responsible for detecting and eliminating RNA intermediates that lack integrity or functionality [1,2,3]. Such transcripts can arise due to deleterious or genomic frameshift mutations or inappropriate processing, and the subsequent failure to produce functional proteins may result in disease.

If an mRNA is devoid of a stop codon (for instance, in the case of truncation or premature 3′-end cleavage and polyadenylation), it will cause the ribosome to progress to its 3′ extremity and stall. Such aberrant mRNAs are rapidly degraded through a process termed non-stop decay [4,5,6]. In contrast, the presence of stable structures or damaged nucleotides within an open reading frame (ORF) can impede ribosome progression, resulting into ribosome stalling upstream of the stop codon. In this case, the transcript is targeted to the degradation by the no-go decay pathway [7,8].

The Nonsense-Mediated mRNA Decay (NMD) is another quality control pathway targeting transcripts that terminate translation prematurely [9,10], such as mRNAs harboring a premature termination codon (PTC) within the ORF [11], as well as PTC-less mRNAs displaying long 3′ untranslated regions (UTRs) [12,13] or short upstream ORFs [13,14,15,16]. The NMD-targeted mRNAs are rapidly degraded [17,18], thus preventing the production of truncated, possibly deleterious proteins [19,20,21].

Here, we review the different mechanisms of NMD activation in yeast and mammals. We discuss the recent evidence showing that NMD also targets and regulates the expression of long non-coding (lnc)RNAs, including antisense (as)lncRNAs in yeast, indicating that translation is part of the metabolism of transcripts initially thought to be devoid of coding potential. Supporting this idea, we describe several examples of functional micropeptides produced from small (sm)ORFs of lncRNAs. We propose that NMD and translation take part in the metabolism of lncRNAs, regulating their expression and providing the opportunity to produce micropeptides which might have a role in the cellular response to environmental changes.

## 2. Discovery, Conservation and Functions of NMD

NMD is a translation-dependent RNA decay pathway [22,23,24], which has been evolutionarily conserved [10,25]. It was originally discovered in the budding yeast *Saccharomyces cerevisiae* by Losson and Lacroute, when they observed that the presence of nonsense mutations reduces the level of a mutant mRNA without affecting its synthesis rate [26]. It was discovered afterwards in humans in the context of β^0^-thalassemia, where it was observed that β-globin mRNAs levels dramatically decrease when carrying nonsense mutations [27,28].

Upstream frameshift proteins (Upfs) 1, 2 and 3 constitute the conserved core components of NMD [29] and were initially identified in *S. cerevisiae* [30,31,32].

Upf1 is a monomeric, highly regulated superfamily 1 helicase. Its ATPase and helicase activities are essential for NMD [33,34]. Upf1 has the ability to translocate slowly but with high processivity on nucleic acids and to unwind long double-stranded (ds)RNA structures [35]. Upf2 is the second core NMD factor and functions as a bridge between Upf1 and Upf3 [36,37,38]. Its interaction with Upf1 is a prerequisite for the phosphorylation of Upf1 [39]. However, NMD can be activated independently of Upf2 [40,41]. Upf3 is the least conserved of the three core NMD factors [42]. Vertebrates have two Upf3 paralogs, Upf3A and Upf3B [37]; in human cells, Upf3B seems to be the main contributor to NMD [43]. Like Upf2, Upf3 stimulates the ATPase and helicase activity of Upf1 in vitro [36]. In metazoans, NMD requires four additional factors: Smg1, Smg5, Smg6, and Smg7 [10,20,44,45]. Interestingly, there is a correlation between the organism complexity and the dependency on NMD; while Upf1 is essential in *Arabidopsis*, *Drosophila* and vertebrates [46,47,48,49], NMD-deficient mutants in yeast and *C. elegans* are viable [20,30,32,50].

At present, it has become clear that the mRNA quality control represents only one face of the multiple functions of NMD [51,52,53,54,55]. In yeast, almost half of protein-coding genes can generate NMD-sensitive mRNA isoforms, including truncated mRNAs for which transcription initiation occurs downstream of the canonical translation initiation site [56]. NMD also targets intron-containing pre-mRNAs that have escaped splicing and were exported to the cytoplasm [14]. In addition, NMD regulates 3–10% of physiological, non-mutated mRNAs in yeast, *Drosophila* and humans, including mRNAs with small upstream ORFs [13,14,15,16,57,58,59,60,61], long 3′ UTRs [12,13], as well as mRNAs displaying low translational efficiency and average codon optimality [14]; considered together, NMD provides a significant contribution to the post-transcriptional regulation of gene expression [55].

Numerous physiological processes rely on the capacity of the cell to adjust NMD activity at global and/or transcript specific levels. NMD factors are essential for embryonic development in vertebrates, as disrupted expression of core NMD factors confers lethality at an early embryonic stage [47,62]. NMD is also crucial for the maintenance of hematopoietic stem and progenitor cells [62], the maturation of T cells [62], as well as for liver development, function and regeneration in mice [63]. Furthermore, NMD is important for the response to multiple stresses [64,65,66], being itself regulated in response to stresses such as hypoxia [67] and amino acid deprivation [68]. In fact, many stress-related mRNAs are targeted by NMD under normal physiological conditions but are stabilized upon stress, due to the inhibition of NMD activity [69]. However, as Upf1 is also involved in diverse RNA decay pathways distinct from NMD, including staufen-mediated mRNA decay, replication-dependent histone mRNA decay, glucocorticoid receptor-mediated mRNA decay, regnase 1-mediated mRNA decay, and tudor-staphylococcal/micrococcal-like nuclease-mediated microRNA decay [70], it remains possible that some of the phenotypes associated with mutants of Upf1 do not reflect the loss of NMD per se. Finally, seven NMD factors (Upf1, Upf2, Upf3B, Smg1, Smg5, Smg6, and Smg7) have been found to be NMD targets in mouse and human cells, revealing the existence of a regulatory feedback network between NMD factors, which is critical for the maintenance of physiological NMD activity and RNA homeostasis [71].

## 3. Molecular Bases of NMD Activation

In many organisms, NMD has been coupled to pre-mRNA splicing [24,72,73,74,75,76,77]. The Exon Junction complex (EJC) is deposited by the spliceosome at the level of the junction between two exons [78], and it is normally removed from the coding regions by the translating ribosomes [79]. The EJC is formed around four core components: the DEAD-box RNA helicase eIF4A3, MLN51, and the Magoh/Y14 heterodimer [80]. The presence of an EJC downstream of a stop codon is recognized as an abnormal situation and enhances the association and activity of Upf1 [81]. In the EJC-enhanced NMD model (Figure 1a), premature translation termination involves the SURF (Smg1–Upf1–eRF1–eRF3) complex, which consists of the Smg1 kinase, Upf1 and the eukaryotic release factors eRF1 and eRF3, and associates with the ribosome stalled at the PTC [82]. Upf2 and Upf3 are then recruited to SURF via the proximal EJC, leading to the formation of the DECID (DECay InDucing) complex [82]. The interaction with Upf2 induces a conformational change in Upf1, allowing its phosphorylation by Smg1 and its activation [82]. The activated Upf1 recruits the Smg6 endonuclease [83] and the Smg5–7 heterodimer [84], which in turn activates RNA deadenylation and decapping. In addition, phosphorylated Upf1 also prevents new translation initiation events by interacting with the translation initiation factor eIF3, inhibiting the formation of a competent translation initiation complex [85]. Finally, protein phosphatase 2 (PP2A) dephosphorylates Upf1, allowing it to return to its unphosphorylated state for another NMD cycle [84].

In addition to the EJC-enhanced NMD, examples of EJC-independent NMD have been described in human cells [86,87], as well as in fission yeast [88], *C. elegans* [89], *Drosophila* [90] and plants [72], all of which have orthologs of EJC factors. In contrast, in *S. cerevisiae*, not only is the proportion of intron-containing genes low (4%) [91], but EJC factors are absent, with the exception of eIF4A3 (Fal1), which acts in pre-rRNA processing in yeast [92].

The EJC-independent NMD targets RNAs with extended 3′ UTR but lacking EJC downstream of the translation termination codon [77,87,93,94,95,96]. Indeed, RNAs where long EJC-free sequences are inserted downstream of a stop codon show reduced levels due to accelerated degradation by NMD [93,95]. This EJC-independent NMD might be a vestige of an ancestral NMD mechanism associated with an abnormally long 3′ UTR, referred to as “faux 3′ UTR”, which is still present in *S. cerevisiae* [97]. In this model, a compromised interaction between the polyadenylate-binding protein Pab1 and the prematurely terminating ribosome results in less efficient termination and enhanced interaction between Upf1 and eRF1/eRF3, triggering NMD (Figure 1b). In this context, a recent proteomics-based analysis in yeast characterized the composition of two distinct NMD complexes associated with Upf1 named Upf1-23 (Upf1, Upf2, Upf3) and Upf1-decapping [98]. The latter contained the decapping enzyme Dcp2 and its co-factor Dcp1, the decapping activator Ebs3, and two poorly characterized proteins, Nmd4 and Ebs1. The Upf1-23 complex is recruited and assembled on the RNA substrate, and then a complete re-organization leads to the replacement of the Upf2/3 heterodimer by Nmd4, Ebs1, Dcp2 and its co-factors (Figure 1b). Nmd4 and Ebs1 are accessory factors for NMD and could be functional homologues of human Smg6 and Smg5/7, respectively [98,99]. The discovery of these new factors in yeast suggests that NMD mechanisms could be more conserved than previously thought. However, how the switch from the “Upf1-23” to an “Upf1-decapping” complex occurs remains unclear.

The polyadenylate-binding protein 1 (PABPC1 in mammals, Pab1 in yeast) is known to stimulate translation termination efficiency by recruiting the release factors to the ribosome [100]. A long distance between the PTC and Pab1/PABPC1 triggers NMD in all studied species [87,93,95,101,102], while tethering it close to the PTC suppresses the NMD sensitivity of the PTC-containing transcripts in yeast [97] and *Drosophila* cells [101]. Mechanistically, it has been proposed that the long 3′ UTR would act by impeding the efficient interaction between Pab1/PABPC1 and eRF1/eRF3, favoring the recruitment of Upf1 by the latter and the formation of a SURF complex at the level of the PTC.

Currently, several questions remain open regarding Upf1 recruitment to the target transcripts. Until recently, the classical view was that Upf1 is recruited at the level of the nonsense codon by the stalled ribosome through an interaction with eRF1/eRF3. However, it has been shown that substrate discrimination by NMD can occur independently of Pab1/PABPC1 or its interaction with eRF3 [103,104], indicating that other features contribute to RNA recognition by NMD. In addition, if Upf1 preferentially binds NMD-targeted transcripts [61,105,106,107], with a marked enrichment in the 3′ UTR [81,108,109,110,111], it is redistributed into the coding sequence upon translation inhibition [109,110,111]. This suggests that Upf1 can bind the RNA independently of translation as well as to NMD non-targets and is pushed away from the coding region by the elongating ribosomes (Figure 2). This means that NMD substrate selection occurs after Upf1 association with the RNA. In this regard, NMD substrate discrimination was shown to rely on a faster dissociation of Upf1 from non-target mRNAs, and this depends on its ATPase activity [106,112]. ATP hydrolysis by Upf1 is also required for ribosome release and recycling and efficient RNA degradation [113,114].

## 4. Long Non-Coding RNAs: An Unexpected Class of NMD Substrates

Unexpectedly, recent transcriptome-wide analyses of RNA binding sites of Upf1 in human and yeast cells revealed that, in addition to mRNAs, Upf1 can also bind lncRNAs [111,115,116].

LncRNAs are a prominent class of transcripts that play important roles in multiple cellular processes, including chromatin modification and regulation of gene expression [117,118,119,120]. They were a priori presumed to be devoid of coding potential [121]. However, this initial assumption has been challenged over recent years by a number of analyses showing that transcripts produced from non-coding regions of the genome, including intergenic regions and sequences antisense to protein-coding genes, associate with the translation machinery in different models, including *S. cerevisiae* [122,123,124,125,126], fission yeast [127,128], plant [129], *Drosophila* [129,130], zebrafish [129,131,132], mouse [129,133] and human cells [129,131,134,135,136]. Thus, not only could the ribosome constitute a default destination for cytoplasmic lncRNAs [136], but the smORFs they carry are likely to be translated into micropeptides [129]. Furthermore, the observation that translation elongation inhibitors results in the stabilization of polysomal lncRNAs in human (K562) cells indicates that translation also determines the degradation of cytoplasmic lncRNAs [136].

In budding and fission yeasts, cytoplasmic lncRNAs are extensively degraded by the 5′-3′ exoribonuclease Xrn1/Exo2 [124,137,138,139]. Inactivation of Xrn1 leads to the stabilization of Xrn1-sensititve Unstable Transcripts (XUTs), the majority of which are antisense to protein-coding genes [124,138,139,140]. Strikingly, in *S. cerevisiae*, 70% of these XUTs are targeted to Xrn1 through NMD [56,124], indicating that most XUTs are translated and that translation constitutes a prerequisite for their degradation. In fact, NMD-sensitive XUTs display ribosome footprints restricted to their 5′ regions, followed by long downstream ribosome-free regions [124]. Conversely, antisense (as)XUTs were found to form dsRNA structures with their paired-sense mRNAs, thus modulating their sensitivity to NMD [124]. This suggests that unless blocked by dsRNA structures, ribosomes could rapidly bind smORFs in the 5′ region of cytoplasmic lncRNAs (Figure 3). The detection of a long 3′ UTR would trigger NMD, leading to the decapping of the transcript and its degradation by Xrn1. Alternatively, but not exclusively, dsRNA could also interfere with the recruitment of NMD factors to asXUTs. Given the current view of Upf1 binding to the RNA (Figure 2) and the observation that Upf1 physically interacts with yeast lncRNAs [115], we propose that Upf1 binds XUTs in a promiscuous manner, independently of translation, regardless of whether or not the transcript will be targeted to Xrn1 through NMD. As proposed for mRNAs, Upf1 would be displaced from the smORF of XUTs by the translating ribosomes and would accumulate on the 3′ UTR. Since NMD-sensitive XUTs are globally longer than NMD-insensitive ones [124], we speculate that longer XUTs carry longer 3′ UTRs, which will be more likely to impede the interaction between eRF1/eRF3 and Pab1. Instead, this situation would favor the Upf1–eRF1/eRF3 interaction, enclosing the XUT as an NMD target through a mechanism similar to the “faux 3′ UTR” model. Supporting this idea, XUTs are poly-adenylated [124,139], and their poly(A) tail is likely to be bound by Pab1.

We speculate that cytoplasmic smORFs-bearing lncRNAs, reminiscent of the yeast NMD-sensitive XUTs, could be targeted by NMD in other eukaryotic cells. Consistent with this idea, NMD inactivation in mouse embryonic stem cells results in the stabilization of a subset of lncRNAs [123]. Moreover, the levels of lncRNAs, including Natural Antisense Transcripts, are also modulated by NMD in *Arabidopsis* [141]. Further support comes from the observation that the *growth arrest-specific 5* (*GAS5*) lncRNA is targeted by NMD and accumulates in Upf1-depleted human cells [142]. 

## 5. Functional Importance of NMD and Translation in lncRNA Metabolism

NMD could be seen as an additional pathway contributing to the clearance of unproductive and potentially harmful spurious non-coding transcripts. However, we believe that this view is too reductive and that there might be more behind the involvement of NMD in the metabolism of lncRNAs (Figure 4). For example, as NMD is a cytoplasmic process, it could ensure that regulatory lncRNAs exhibit their functions exclusively in the nucleus by limiting their accumulation outside the nucleus. In addition, NMD could limit the accumulation of nonfunctional and potentially deleterious peptides from cytoplasmic lncRNAs during the de novo gene birth [125,126,143]. Additionally, the peptides produced from NMD-sensitive lncRNAs could be functional and important, despite their low levels. Even if this remains completely speculative for NMD-sensitive lncRNAs, we note that antigens of the MHC class I pathway are produced from PTC-containing mRNA [144], raising the question of how this process might be generalized for “cryptic” lncRNAs.

NMD could also specifically modulate the levels of regulatory lncRNAs. The apoptotic lncRNA *GAS5* has been proposed to act in an NMD-based circuit, which is critical in response to serum starvation [142]. In normal conditions, NMD restricts the constitutive *GAS5* expression to low levels. However, in stress conditions associated with NMD inhibition (such as serum starvation), *GAS5* expression is up-regulated and binds the glucocorticoid receptor, perturbing its function as a transcription activator in the anti-apoptotic program [142].

More globally, by targeting regulatory cytoplasmic aslncRNAs, NMD could contribute to regulate gene expression. For instance, stabilization of subsets of Xrn1-sensitive aslncRNAs, most of which are NMD-sensitive [124,128], correlates with the transcriptional attenuation of the paired-sense genes, in budding and fission yeasts [137,139]. Interestingly, two independent studies in zebrafish embryos reported that NMD factors cycle to the nucleus to trigger transcriptional adaptation of genes with a sequence complementarity to the PTC-containing RNA in a mechanism called genetic compensation [145,146].

Finally, coupling translation to aslncRNA degradation via NMD could be important for cell recovery upon translation inhibitory stress. In such a condition, NMD-sensitive aslncRNAs are expected to be stabilized and form duplexes with their paired-sense mRNAs. By analogy with the protective effect on the aslncRNA [124], this interaction could also prevent the degradation of the mRNA partner, since local dsRNA formation correlates with higher mRNA stability [147]. After stress, the protected sense/as transcripts would be rapidly released upon the action of RNA helicases, thereby providing a pool of mRNAs in the cytoplasm that can be translated, while NMD-sensitive aslncRNAs would be rapidly degraded.

Together, the observations reported above support the idea that NMD is able to target lncRNAs, and that this might be important for the maintenance of RNA homeostasis, the regulation of gene expression, and for a robust response to several stress conditions. It also challenges the initial assumption that such transcripts are devoid of coding potential.

## 6. Insight into the Coding Potential of “Non-Coding” Transcripts

The accumulating evidence that cytoplasmic lncRNAs interact with the translation machinery raises the question of their coding potential. Numerous methods have been developed to assess this possibility [148,149,150]. Additionally, a growing body of experimental data indicate that “non-coding” RNAs can indeed be translated [151], including not only lncRNAs but also circular RNAs [152,153,154] and primary microRNAs transcripts [155,156]; moreover, these translation events can produce functional peptides [134,151,157,158,159].

“Non-coding” RNAs contain one or more smORFs that can be translated into micropeptides (i.e., peptides not exceeding 100 amino acids in length.) Previously, such smORFs were ignored as the traditional gene annotation process filtered out ORFs shorter than 100 codons, considering them as noise or false positives. However, as ribosome profiling techniques and proteomics are growing in popularity and increasing in sensitivity, accuracy and efficiency, it is becoming clear that at least a fraction of short ribosome-bound sequences of (l)ncRNAs represent genuine smORFs. 

Importantly, a recent work from Weissman’s lab provided a catalog of smORFs and functional peptides derived from human lncRNAs, which included the identification of >800 novel lncRNA-associated smORFs and the observation that, for 91 of them, CRISPR-mediated knockout of the smORF resulted in a growth phenotype [134], indicating that the corresponding peptides are important for cell growth. Other studies previously showed that lncRNA-derived micropeptides are involved in the regulation of RNA decapping [160], in embryonic development [161,162], in muscle development [163,164,165], regeneration [166,167] or contraction [168,169,170], and in tumor development [154,171,172] (see Table 1).

Mechanistically, if global information about the mode of action of lncRNA-derived peptides is still lacking, pioneer studies revealed that they can act by binding other proteins and regulate their activity [162,168,169,176], or as signaling pathway molecules [177]. We anticipate that future works will reveal additional modes of action.

In the light of the observations that micropeptides produced from lncRNAs can be biologically important, it is tempting to speculate that aberrant expression of endogenous lncRNA-derived peptides could be associated with diseases, including cancer [154,171]. In addition to providing a new perspective on pathogenicity, lncRNA-derived peptides could also constitute promising targets for targeted therapy [178], including tumor immunotherapy [179]. In this respect, it is interesting to note that a recent characterization of different murine cell lines and cancer patient samples showed that non-coding regions constitute the major source of tumor specific neo-antigens [179], which could be pivotal for the development of future immunological treatments and cancer vaccines [180]. 

## 7. Conclusions

Today, NMD is extending far beyond its original definition assigning it only to the clearance of aberrant “nonsense” transcripts. The current research has revealed that it provides potent means to regulate the expression of many mRNAs and lncRNAs, as well as contributes to the establishment of suitable cellular responses to environmental changes, including adaptation, differentiation or apoptosis. The accumulating biochemical and transcriptomic evidence showing that NMD targets lncRNAs implores us to reconsider the idea that lncRNAs are devoid of coding potential, and challenges us to address how translation of smORFs could not only affect their stability, but also could be used to produce functional micropeptides. Revealing the possibility for a “dark peptidome” to arise from the “dark non-coding side of the genome” (i.e., “the dark side of the dark matter”) is one of the challenges in the RNA field for the coming years and will open exciting perspectives regarding the roles of lncRNA-derived peptides.

## Figures and Tables

**Figure 1 ncrna-07-00044-f001:**
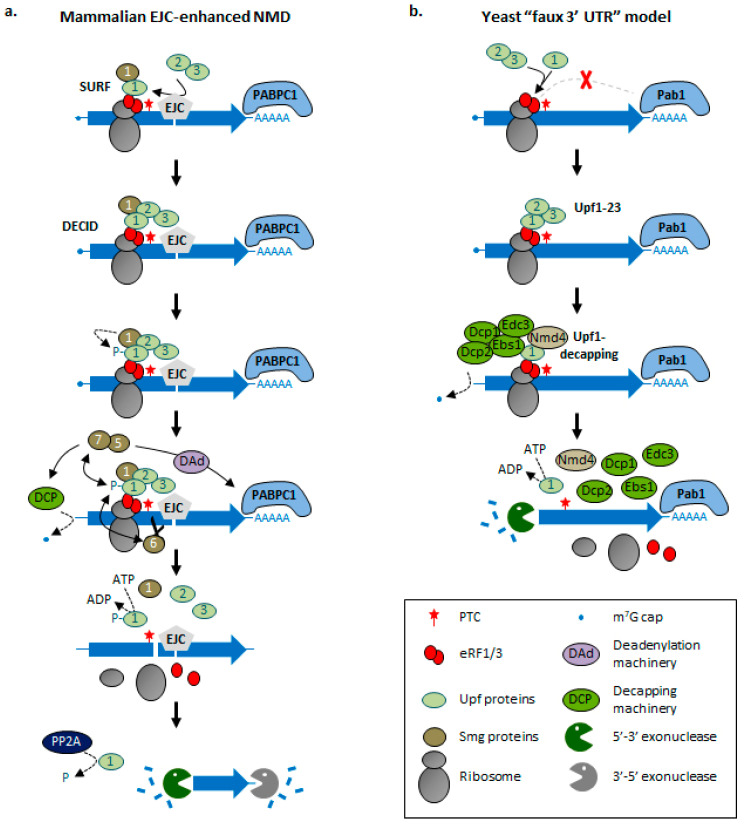
Models of NMD activation mechanisms in mammals and yeast. (**a**) Mammalian EJC-enhanced NMD. When an EJC remains bound to the RNA downstream of a termination codon, translation termination is inefficient as the EJC interferes with the interaction between PABPC1 and the eukaryotic release factors (eRF1/eRF3). Instead, a SURF complex (Smg1–Upf1–eRF1–eRF3) forms at the level of the PTC. Upf2 and Upf3 are then recruited by the downstream EJC and associate with SURF to form the decay-inducing (DECID) complex. Smg1 phosphorylates Upf1 (P), activating it. Phosphorylated Upf1 promotes RNA decay via Smg6-dependent endonucleolytic cleavage and the Smg5–Smg7-dependent triggering of mRNA deadenylation and decapping. ATP hydrolysis by Upf1 allows the dissociation of the termination complex and the release of the transcript, which can be degraded. Upf1 dephosphorylation by protein phosphatase 2A (PP2A) allows it to return to a dephosphorylated state. The coding region and the UTRs of the mRNA are represented as a large blue arrow and thin blue lines, respectively. See the key for the other symbols. (**b**) “Faux” 3′ UTR model in yeast. A long 3′ UTR results in inefficient translation termination and Upf1 interaction with eRF1/eRF3, promoting the formation of the Upf1-23 complex (Upf1, Upf2, Upf3) at the level of the terminating ribosome. The Upf2-Upf3 heterodimer is then replaced by Nmd4, Ebs1, the decapping enzyme Dcp2 and its co-factors Dcp1 and Edc3 in the Upf1-decapping complex, leading to RNA decapping. ATP hydrolysis by Upf1 promotes the disassembly of the mRNA/ribosome/Upf1-decapping complex, leading to the release of the transcript which can finally be degraded by Xrn1.

**Figure 2 ncrna-07-00044-f002:**
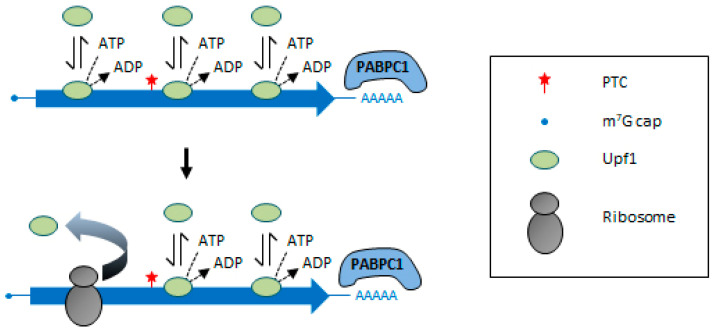
Model of translation-dependent displacement of Upf1 from the mRNA coding region. Upf1 binds promiscuously to any accessible RNA (including NMD non-targets), independently of translation. ATP hydrolysis promotes Upf1 dissociation from non-target RNA regions. Upf1 is also displaced from the coding region by the translating ribosome. This model implies that NMD substrate selection occurs after Upf1 associates with the RNA.

**Figure 3 ncrna-07-00044-f003:**
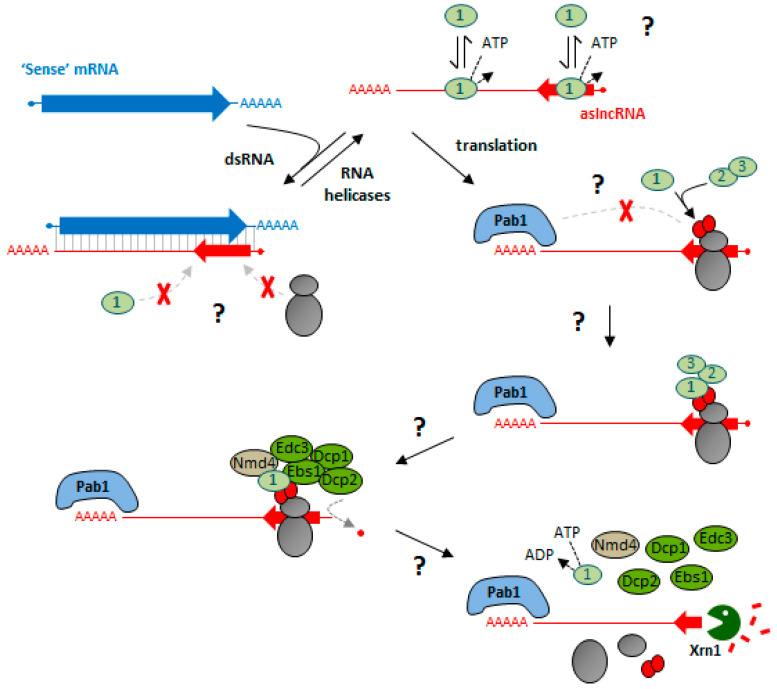
Model of yeast cytoplasmic aslncRNA degradation by NMD. Once in the cytoplasm, an aslncRNA (red) would rapidly be bound by ribosomes, unless in a dsRNA structure with its paired-sense mRNA (blue). This dsRNA might also interfere with Upf1 binding to the aslncRNAs and could be removed by the action of RNA helicases. The detection of a long 3′ UTR would trigger NMD by a mechanism similar to the “faux 3′ UTR”. A Upf1-23 complex would form at the level of the termination codon thanks to the interaction between Upf1 and eRF1/eRF3. The subsequent formation of the Upf1-decapping complex would lead to the decapping of the aslncRNA by Dcp2. Upon ATP-dependent disassembly of the complex, the decapped aslncRNA is degraded by Xrn1. The mRNA and the aslncRNA are represented in blue and red, respectively. Large arrows and thin lines represent the coding regions and the UTRs, respectively. The ribosome and NMD/decapping factors are represented as in Figure 1.

**Figure 4 ncrna-07-00044-f004:**
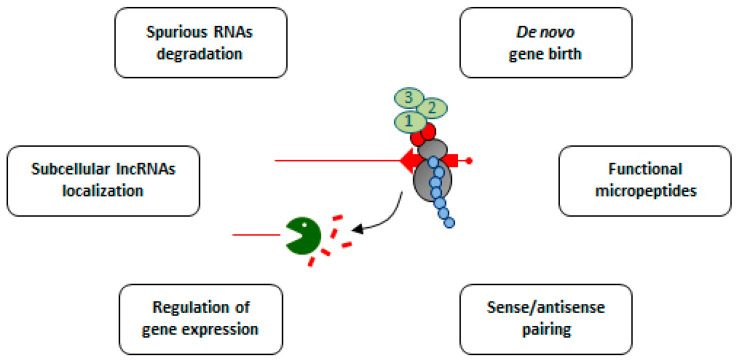
Possible roles of translation and NMD in the metabolism of (as)lncRNAs. Schematic representation of the functional importance of NMD and translation in lncRNAs metabolism (see main text for details). The chain of blue balls represents the micropeptide produced upon the translation of the smORF (red arrow) of the (as)lncRNA. The ribosome, the (as)lncRNA and the NMD/decay factors are represented as above.

**Table 1 ncrna-07-00044-t001:** Examples of functional lncRNA-derived micropeptides.

Micropeptide	Species	Target	Function(s)	Ref.
NoBoDy	Human	mRNA decapping factors	Regulation of mRNA turnover and P-body numbers	[160]
CASIMO1	Human	Squalene epoxidase	Carcinogenesis; cell lipid homeostasis	[171]
PINT87aa	Human	Polymerase associated factor complex (PAF1c)	Oncogene transcriptional inhibition; tumor suppressive effect	[154]
HOXB-AS3	Human	hnRNP A1 splicing factor	Colon cancer growth suppression	[173]
RBRP	Human	m^6^A reader IGF2BP1	Regulation of m^6^A recognition by IGF2BP1 on c-Myc mRNA; tumorigenesis	[172]
Minion/ Myomixer	Human, mouse	Unknown	Myoblast fusion; muscle formation and development	[163,164,167]
SPAR	Human, mouse	Lysosomal v-ATPase	Regulation of mTORC1 signaling pathway; muscle regeneration	[166]
TUG1-BOAT	Human, mouse	Unknown	Unknown; alters mitochondrial membrane potential when overexpressed	[174]
Mtln	Human, mouse	Cardiolipin	Increase of mitochondrial functions	[175]
DWORF	Mouse	SERCA	SERCA (sarcoplasmic reticulum Ca^2+^-ATPase) activation	[168,169]
MLN	Mouse	SERCA	SERCA inhibition	[176]
Toddler	Zebrafish	Unknown	Promoting cell migration during embryogenesis	[174]
Pri	*Drosophila*	Ubr3 E3 ubiquitin ligase	Proteasome-dependent processing of the developmental Svb transcription factor	[162]
Scl	*Drosophila*	Ca-P60A SERCA	Calcium transport regulation	[170]

## Data Availability

Not applicable.

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
