# Peer review of "From Yeast to Mammals, the Nonsense-Mediated mRNA Decay as a Master Regulator of Long Non-Coding RNAs Functional Trajectory"

_ncrna, 2021, doi:10.3390/ncrna7030044_

Round 1

Reviewer 1 Report

Over the past 10 years or so, there has been evidence accumulating that most lncRNAs that make it to the cytoplasm do actually engage with ribosomes (i.e. they are not strictly “non-coding”), leading to the production of micropeptides and in most cases render these lncRNAs to be degraded by NMD. This manuscript is to my knowledge the first that systematically and comprehensibly summarizes this evidence and reviews these papers. In that perspective, I find the manuscript important and timely. The authors are experts on yeast lncRNAs and this part is therefore, unsurprisingly, very well written and all information is correct, while the parts on NMD are not always entirely accurate and need some amendments (see below).

Specific points to address:

  • In order to confirm with the rest of the literature, the authors should spell out NMD as “Nonsense-mediated mRNA decay” instead of “Nonsense-mediated decay”. If the authors intentionally left out the word “mRNA” because in their review they want to emphasize that also lncRNAs can be targeted by NMD, they should in the beginning explain this clearly.
  • 2, lines 57-59: The first description of NMD in human cells the following paper: Maquat et al., Cell 1981. The cited paper by Baserga et al. was published 7 years later.
  • 1a: Given that SMG6-induced endonucleolytic cleavage is the predominant way of triggering NMD in Drosophila and mammals, it is odd that only the minor pathway (SMG5/7-mediated deadenylation) is depicted and SMG6 is not shown. The authors should adjust their drawing of the model.
  • In several places it is mentioned that NMD occurs “during the pioneer round of translation”. While this model has been promoted strongly by Lynne Maquat over the past years, it is clearly disproved and wrong. Not only is the faux 3’ UTR incompatibly with a restriction of NMD to the first ribosome translating it, it has also been shown that eIF4E-bound mRNAs are still a substrate for NMD (Durand & Lykke-Andersen, NSMB 2013; Rufener & Mühlemann, NSMB 2013) and a recent live cell single-molecule study by the Tanenbaum lab indicated that on the classic b-globin ter39 NMD reporter transcript, NMD is triggered on average by every 10th ribosome translating it (Hoek et al., Mol Cell 2019). I would therefore suggest to remove references to the pioneer round model, since these controversial mechanistic issues about NMD are not the focus of this manuscript and hence only distract from thew main message, namely that many lncRNAs are targeted for decay by NMD.

Reviewer 2 Report

In this Review, Andjus et al. provide a review on literature about basic mechanisms of NMD and the interplay between NMD and lncRNAs, the latter being more speculative (which is fine). The authors mention the main points of NMD although there are certain fragments of text that are a little difficult to read. The section on lncRNAs and translation is interesting, but the authors should rephrase certain things, and where there are certain statements provide enough support from previous literature. In particular, my suggestions are:

Introduction

In the Introduction I would clearly state that there are several forms of decay rather than starting with the position of the stop codons. NMD is also triggered by long 3’UTRs for example so it is not all stop codon related.

Discovery

Line 60: I believe the authors mean Upstream frameshift 1 (Upf1).

Line 67: There is evidence the UPF1 can function without the need of UPF2 (Aznarez et al., 2018 Cell Reports) please add this information.

Line 71: what about plant NMD? Mutants in UPF1 are lethal.

Please check spelling e.g. NMD rather than ‘the NMD’.

Line 77: ‘canonical’ rather than ‘normal’?

Lines 78-79: these lines seem to contradict your lines 119-123. Splicing is mostly absent in yeast, what is the proportion of intron-containing mRNAs? Yeast is precisely mentioned as an example of organism where NMD occurs independent of splicing. Please merge both concepts so that they are not antagonising.

Line 81: ‘normal looking mRNAs’ please define using other phrasing.

Line 94: please give examples of NMD factors involved in cellular pathways that are not related to NMD. What factors, what pathways?

Molecular basis of NMD activation

Line 101: there is a typo: basis, not bases

Line 104: please define the core components of the EJC.

Please start with the different models and then drill down on each one.

Use all the letters of acronyms (eRF1 and eRF3, line 108).

The SURF complex does not contain a ribosome, it is associated with one.

Line 124: please mention the effect of 3’UTR length on NMD.

Line 130: Pab1 is Polyadenylate binding protein. In this model, is the ribosome prematurely terminating, or is there a delay in the termination, and binding of UPF1 that triggers NMD?

Line 143: there are several PABPs please specify, do the authors mean PABPC1? Or if they mean PABPs in general referring to several species please do state this clearly. In Figure 1 PABPC1 is clearly shown.

With regards to UPF1 it is thought to bind the entire transcriptome so there is also the possibility of ‘indiscriminate’ binding of UPF1 that can also trigger NMD in for example 3UTR EJC independent NMD. Please include a legend in Figure 2 similar to Figure 1.

Long non-coding RNAs

Is there a role for lncRNAs that does not rely on them being translated into small peptides – i.e. as microRNAs can also appear associated with ribosomes?

The authors mention the pioneer round of translation on line 222 and I believe there should be a few lines dedicated in the previous section given 1) the dependency of NMD on ribosome scanning (not necessarily translation) 2) the controversy around CBC vs eIF4E cap binding and NMD (Maquat and Lykke-Andersen groups, predominantly).

Line 245: what does in fine mean please.

NMD and translation in lncRNAs

I do not think that the field shares the view of lncRNAs being a non-functional product. There is plenty of literature showing otherwise, please remove or rephrase this.

Figure 4 should say ‘possible roles’. I am also not convinced about the title, the authors are mixing concepts of lncRNas being or not translated, and NMD in the middle of that. Please review evidence of lncRNAs not associated with ribosomes that are NMD targets, this would perhaps focus this section a little more in those that are present in ribosomes vs that not ribosomes. Likewise, there is a debate about micropeptides from lncRNas making those lncRNas not truly ‘non-coding’. If they have coding potential, then they surely cannot be lncRNAs, by definition. They should be called something else.

Insight into coding potential of lncRNAs

This section is totally different from NMD. I can tell that the authors are very excited about it, which is fine, but they should perhaps merge with the previous one, and end in NMD. The study by Weissman with growth changing, that does not indicate that those peptides are important for cell growth, in my opinion. LncRNAs can function as RNAs and exert their functions in that way.
